# HSA-ZW800-PEG for Enhanced Optophysical Stability and Tumor Targeting

**DOI:** 10.3390/ijms25010559

**Published:** 2023-12-31

**Authors:** Paul Jang, Jinhui Ser, Kevin Cardenas, Hajin Joanne Kim, Morgan Hickey, Jiseon Jang, Jason Gladstone, Aisha Bailey, Jason Dinh, Vy Nguyen, Emma DeMarco, Surbhi Srinivas, Homan Kang, Satoshi Kashiwagi, Kai Bao, Atsushi Yamashita, Hak Soo Choi

**Affiliations:** 1Gordon Center for Medical Imaging, Department of Radiology, Massachusetts General Hospital and Harvard Medical School, Boston, MA 02119, USA; pjang1@mgh.harvard.edu (P.J.); jser1@mgh.harvard.edu (J.S.);; 2School of Materials Science & Engineering, Chonnam National University, Gwangju 61186, Republic of Korea

**Keywords:** NIR imaging, photostability, tumor targeting, human serum albumin, enhanced permeability and retention

## Abstract

Small molecule fluorophores often face challenges such as short blood half-life, limited physicochemical and optical stability, and poor pharmacokinetics. To overcome these limitations, we conjugated the zwitterionic near-infrared fluorophore ZW800-PEG to human serum albumin (HSA), creating HSA-ZW800-PEG. This conjugation notably improves chemical, physical, and optical stability under physiological conditions, addressing issues commonly encountered with small molecules in biological applications. Additionally, the high molecular weight and extinction coefficient of HSA-ZW800-PEG enhances biodistribution and tumor targeting through the enhanced permeability and retention effect. The unique distribution and elimination dynamics, along with the significantly extended blood half-life of HSA-ZW800-PEG, contribute to improved tumor targetability in both subcutaneous and orthotopic xenograft tumor-bearing animal models. This modification not only influences the pharmacokinetic profile, affecting retention time and clearance patterns, but also enhances bioavailability for targeting tissues. Our study guides further development and optimization of targeted imaging agents and drug-delivery systems.

## 1. Introduction

Optical fluorescence imaging has been utilized for monitoring biological activities, offering unprecedented insights into cellular processes as well as the diagnosis and treatment of human diseases [1,2,3]. Near-infrared (NIR) fluorescence, spanning the range of 650–1700 nm, presents a distinct advantage over visible fluorescence detection [4,5]. Its larger dynamic range and diminished background fluorescence, attributed to reduced tissue absorption, scattering, and autofluorescence, make it an invaluable tool for sophisticated imaging techniques [6,7].

Among many commercially available NIR fluorophores, polymethine cyanine fluorophores stand out for their exceptional sensitivity in biological tissues, driven by high extinction coefficients and quantum yields in aqueous solvents [8,9]. Notably, indocyanine green (ICG) has been widely employed in clinical settings, particularly for tumor targeting [10,11] and metastatic lymph node mapping [12,13,14]. ICG is hindered by challenges such as an extremely short blood half-life (<5 min), poor photostability and solubility, and a propensity for aggregation and quenching [15,16]. Moreover, its lack of conjugatable sites limits its broader clinical utility. In our pursuit of more precise and targeted imaging, we previously synthesized zwitterionic heptamethine indocyanine fluorophores, including ZW800-1A [17], ZW800-1C [18], and ZW800-PEG [19], showcasing remarkable optical and physicochemical features such as high molar extinction coefficients, quantum yields, water solubility, and stability [20]. Despite these attributes, these small molecule fluorophores lacked specificity for tumoral tissue [21].

To overcome these limitations, significant efforts have been invested in conjugating NIR fluorophores to various targeting ligands, ranging from small molecule drugs to peptides, proteins, and antibodies [22,23,24,25]. Human serum albumin (HSA), a key biomolecule, has emerged as a promising candidate for enhancing the specificity of targeting within tissues [26,27,28]. Constituting about 60% of total proteins, HSA is the most abundant functional protein in the intravascular and extracellular space [29,30]. With a molecular mass of 66 kDa and a hydrodynamic diameter approximately seven times larger than a small molecule fluorophore [31], HSA exhibits unique physicochemical and pharmacokinetic properties as well as physiological functions. It circulates in the lymphatic system around 28 times during its lifetime, interacting with receptors and boasting a circulating half-life of approximately 19 days [32,33].

ICG can easily adsorb to HSA noncovalently, resulting in the formation of the ICG:HSA complex, offering improved pharmacokinetics, optical stability, and delivery efficiency to the target tissue [34]. However, challenges persist in comprehending the biodistribution and clearance mechanisms, especially concerning unconjugated ICG and its metabolized forms [35]. Notably, beyond a 1:1 molar ratio of ICG to HSA, the improvement in total fluorescence yield becomes saturated, potentially allowing dissociated ICG to travel freely in the body [36,37]. The binding pocket of HSA can also be obstructed by ICG, leading to quick dissociation of ICG after passage through the liver [38]. 

In this context, we engineered HSA-ZW800-PEG by covalently conjugating the highly water-soluble and zwitterionic ZW800-PEG fluorophore onto the HSA surface. This innovative engineering aims to enhance optophysical stability and tumor targetability. As depicted in Figure 1a, HSA can traverse tumor vessels via abnormal leaky junctions (the enhanced permeability and retention (EPR) effect; passive targeting) or interact with the caveolin-1 receptor (active targeting), positioning HSA-ZW800-PEG as a significant candidate for robust tumor targeting compared to noncovalently adsorbed ICG:HSA.

## 2. Results

### 2.1. Synthesis and Optical Characterization of HSA-ZW800-PEG

The 800 nm emitting zwitterionic NIR fluorophore ZW800-PEG [19], along with its NHS ester form, was synthesized as reported previously [20]. In preparation for preclinical and prospective clinical investigations, the imperative was to generate substantial quantities with high yield and purity (>95%). We described the detailed synthesis of ZW800-PEG at a 10 g scale using facile and efficient solvent purification [20]. HSA-conjugated with ZW800-PEG (HSA-ZW800-PEG) was prepared by adding ZW800-PEG NHS ester in the nucleophilic HSA in PBS, pH 8.0 (Figure 1b). The reaction mixture was stirred at room temperature for 3 h, followed by purification using gel-filtration chromatography (GFC) connected with Bio-Scale Mini Bio-Gel P-6 Desalting columns (Bio-Rad, Hercules, CA, USA) [39,40,41]. HSA-ZW800-PEG shows a clean, distinct peak around 4.2 min, while ZW800-PEG appears at 9.2 min, confirmed by a photodiode array (PDA) detector at 254, 660, and 760 nm wavelengths (Figure 1c). This result indicates that no unconjugated ZW800-PEG was included in the final product after GFC purification. HSA-ZW800-PEG shows absorbance and fluorescence spectra in the 800 nm channel with a 23 nm stokes shift, identical to ICG, which can be imaged under the current clinically available intraoperative imaging devices (Figure 1d). The labeling ratio of ZW800-PEG to HSA was calculated to be 2.1 using the extinction coefficients of HSA (35,700 M^−1^cm^−1^ at 280 nm) and ZW800-PEG (192,000 M^−1^cm^−1^ at 787 nm, correction factor 0.047 in water). Quantum yields were measured in saline and compared with ICG in DMSO [19]. The molecular weight (MW) and hydrodynamic diameter (HD) of HSA-ZW800-PEG were improved by about 7-fold compared to those of ZW800-PEG (Figure 1e).

### 2.2. In Vitro Optophysical Stability

Optical and physicochemical stability is a prerequisite of a targeted contrast agent since the changes in brightness or molecular structure can cause false-positive or false-negative signals [42]. As shown in Figure 2a, HSA-ZW800-PEG exhibited over 90% photostability in saline under the exposure of 760 nm laser diode at 4 mW/cm^2^ for 240 min, while the fluorescence signals of ICG and ZW800-PEG were reduced to 65% and 40%, respectively. The absorbance spectra of HSA-ZW800-PEG were proportional to the concentration up to 25 µM (Figure 2b). Additionally, as depicted in Figure 2c, the fluorescence signals showed a linear increase up to 10 µM, plateauing thereafter. This suggests that the quenching effect commenced at concentrations exceeding 12.5 µM, as evidenced by changes in the absorbance and fluorescence spectra.

### 2.3. In Vivo Biodistribution and Pharmacokinetics

The increased MW and HD, along with the enhanced optophysical stability of HSA-ZW800-PEG, collectively contribute to an improved bioavailability and pharmacokinetic profile. To investigate this effect, 50 nmol of HSA-ZW800-PEG (1 µmol/kg) was injected intravenously in 25 g male CD-1 mice. Intraoperative imaging was conducted 4 h post-injection, encompassing the thoracic cavity and abdominal wall, both before and after surgical opening, and animals were sacrificed to find biodistribution and clearance in major organs (Figure 3a). Remarkably, HSA-ZW800-PEG was observed in the liver sinusoids, bone marrow, and spleen. These regions feature discontinuous endothelium characterized by large-diameter fenestrae without diaphragms, along with a poorly organized underlying basement membrane, facilitating the transcellular flow of large solutes, including HSA [43]. Furthermore, elevated fluorescence signals were detected in the bloodstream, brain vasculatures, and choroid plexus. This phenomenon can be attributed to the increased size of albumin resulting from the extended MW and HD [44]. The resected organs of HSA-ZW800-PEG and ZW800-PEG show clear differences in the heart, lungs, liver, pancreas, and spleen (* *p* < 0.05 or ** *p* < 0.01; Figure 3b). Interestingly, the systemic blood circulation of HSA-ZW800-PEG was found to be approximately 10-fold longer than that of ZW800-PEG (Figure 3c). ZW800-PEG is distributed rapidly into major organs (t_1/2α_ = 2.1 min) and then eliminated quickly from the body (t_1/2β_ = 27.0 min) with a fast plasma clearance rate (0.5 mL/min). In contrast, both the distribution and elimination phases of HSA-ZW800-PEG demonstrated a significant improvement, exhibiting approximately 5-fold (t_1/2α_ = 9.7 min) and 10-fold (t_1/2β_ = 250 min) enhancements, respectively, compared to ZW800-PEG. The area under the curve (AUC) of HSA-ZW800-PEG was 8-fold larger and the clearance was about 28-fold slower, respectively, compared to ZW800-PEG (Figure 3d). Approximately 48% ID was found in the urinary bladder and ureters.

### 2.4. Tumor Targetability and Cytotoxicity Assay

In vitro cancer cell targetability of HSA-ZW800-PEG was determined by fluorescence microscopic observation (Cytation5, BioTek, Winooski, VT, USA) using cultured murine LLC lung cancer cell line (ATCC, Manassas, VA, USA). NIH3T3 fibroblast cell line (ATCC) was used as normal cell control. Overall, HSA-ZW800-PEG showed significantly higher accumulation in the cancer cells compared to the control cells (**** *p* < 0.0001), while zwitterionic ZW800-PEG was similarly taken up by both cell lines (Figure 4a). The enhancement of tumor selectivity with HSA has been reported, which is accompanied by the expression level of receptor and extracellular proteins such as caveolin-1, sialoglycoprotein, and secreted protein acidic and rich in cysteine [22,45]. Our results also clearly indicate that HSA-ZW800-PEG could be less associated with normal cells, thus potentially minimizing cytotoxicity and off-target nonspecific uptake. Then, to assess the involvement of endocytosis, Dyngo 4a, an inhibitor of dynamin-dependent endocytosis (Abcam, Waltham, MA, USA, No. 120689), was added to the cells (30 μM) and incubated for 30 min, followed by treating 100 μM of HSA-ZW800-PEG for 8 h at 37 °C. The morphological images of cells were captured using Cytation5 following two rounds of washing in HBSS solution (Figure 4b). Additionally, we confirmed the cytotoxicity of HSA-ZW800-PEG in NIH3T3 fibroblast cells, proving minimum to no cytotoxicity in the range of 2.5–100 µM concentrations (Figure 4c). 

In vivo tumor targetability was evaluated by injecting 50 nmol (2 μmol/kg) of HSA-ZW800-PEG and 100 nmol (4 μmol/kg) of ZW800-PEG into orthotopic and subcutaneous xenograft LLC tumor-bearing mice, respectively. Since two ZW800-PEG moieties are conjugated on the HSA surface (labeling ratio = 2.1), different injection doses were applied. and the fluorescent signals in tumors and adjacent vasculature were concurrently monitored for up to 24 h post-injection (Figure 4d). Shown are animals sacrificed at 8 h post-injection of ZW800-PEG and 24 h post-injection of HSA-ZW800-PEG. Two different time points were selected and compared due to the difference in the blood half-life values and tumor targetability of the tested agents. 

Remarkably, HSA-ZW800-PEG showed high uptake in the tumoral site, as well as in the bone marrow, compared to ZW800-PEG in the subcutaneous xenograft tumor models (Figure 4d, left). The HSA-ZW800-PEG signals were maintained on the tumor over 4 h post-injection, while ZW800-PEG showed an initial peak during the distribution phase and was gradually eliminated from the tumoral site after 2 h post-injection (* *p* < 0.05; Figure 4e). Interestingly, the lung tumor lesions were successfully identified in the orthotopic lung cancer model injected with HSA-ZW800-PEG (Figure 4d, right), resulting in a significantly higher tumor-to-background ratio (TBR) compared with ZW800-PEG (**** *p* < 0.0001; Figure 4f).

## 3. Discussion

The development of effective theragnostic molecules relies on overcoming critical challenges such as enhancing solubility, stability, and targetability [46]. Our investigation centered on leveraging the unique properties of HSA-conjugated NIR fluorophores to achieve superior biological stability and enhanced tumor targeting. HSA was strategically chosen as the conjugation partner for ZW800-PEG. The resulting conjugate, HSA-ZW800-PEG, was efficiently synthesized using conventional NHS ester chemistry in aqueous solvents. To assess the advancements achieved through HSA introduction, rigorous in vitro and in vivo comparisons were conducted with pure ZW800-PEG.

The conjugation of HSA resulted in notable improvements in both the optical and physicochemical stability of a small molecule fluorophore. After 4 h of irradiation, HSA-ZW800-PEG maintained over 90% of its fluorescence signal, while ZW800-PEG and ICG showed significant fluorescence reductions (Figure 2a). This enhanced stability is a pivotal attribute for sustained imaging efficacy, without causing any false-positive or false-negative signals in the body during the imaging study.

HSA-ZW800-PEG exhibited superior tumor targetability in both in vitro and in vivo models. In cellular binding assays, HSA-ZW800-PEG demonstrated substantial uptake in LLC lung cancer cells compared to NIH3T3 normal fibroblast cells. The overexpression of caveolin-1 receptors on malignant cells and stromal cells associated with neoplasia [47,48,49] actively contributes to tumor targeting facilitated by the introduction of HSA (Figure 1a). The choice of a zwitterionic and hydrophilic fluorophore for conjugation, avoiding the hydrophobic binding pocket of HSA, adds a novel dimension [50]. Importantly, HSA-ZW800-PEG exhibited a negligible affinity for NIH3T3 normal fibroblast cells, indicating minimal to no cytotoxicity associated with nontarget cellular uptake.

Consistent with in vitro findings, robust fluorescence signals were detected in tumor tissues in both subcutaneous and orthotopic lung cancer models. These results substantiate our hypothesis that HSA-ZW800-PEG enhances tumor targetability due to its larger size and extended blood circulation. This aligns with numerous reports on the longer blood half-life and passive targeting of HSA-conjugates through the enhanced permeability and retention (EPR) effect [51,52,53,54,55]. Additionally, the high expression of caveolin-1 in tumor stroma, a significant component of caveolae, has been associated with improved responses to albumin-conjugated drugs in non-small cell lung cancer (NSCLC) patients [20,56]. Similar results were found when Cy5.5 was conjugated on albumin, where tumor targeting was improved but background uptake was also increased [57]. Thus, HSA-ZW800-PEG holds the potential not only for passive targeting but also for active targeting. These pivotal advancements stand to optimize cancer imaging and treatment using NIR fluorophores, paving the way for multifaceted applications in cancer therapeutics.

## 4. Materials and Methods

### 4.1. Synthesis and Purification of HSA-ZW800-PEG

HSA in the sodium caprylate and *N*-acetyltryptophan buffer (Flexbumin 25%, 50 mL) was obtained from Takeda Pharmaceuticals (Deerfield, IL, USA) and recollected by using 10,000 molecular-weight cutoff (MWCO) spin columns (Vivaspin 500, 10 K MWCO) with 1x PBS, pH 8.0 (3 times), followed by lyophilization to give the white solid form of HSA. The NHS ester form of ZW800-PEG was synthesized as previously reported (>95% purity) [19]. To conjugate ZW800-PEG on HSA, 10 molar equiv. of ZW800-PEG NHS ester was added into deprotonated nucleophilic HSA in PBS, pH 8.0. The mixture was stirred gently at room temperature for 3 h with protection from light. To purify the unconjugated fluorophores from HSA-ZW800-PEG, gel-filtration chromatography (GFC) was performed using Bio-Scale Mini Bio-Gel P-6 Desalting columns (Bio-Rad, Hercules, CA, USA), with a flow rate of 2 mL/min. HSA-ZW800-PEG fractions were collected and lyophilized to yield the green solid product. After purification, size exclusion chromatography (SEC) analysis was conducted using an analytical high-performance liquid-chromatography (HPLC) system. The HPLC system consisted of a 1525 binary HPLC pump (Waters, Milford, MA, USA) with an Alliance e2695 separation module, a 2998 PDA detector (212-800 nm), 2489 UV/Visible detector, and an Acquity QDA detector (m/z range: 50-1,239). The column eluent was split into two using a flow splitter (Upchurch Scientific, VWR, Oak Harbor, WA, USA). A portion of the eluent was directed into the PDA equipped with an Xbridge BEH125 SEC (7.8 × 150 mm, 3.5µm, Waters, Milford, MA, USA). The mobile phase was maintained in PBS for 30 min, and the flow rate was set at 0.75 mL/min. The chemical properties of contrast agents were calculated using MarvinSketch (ChemAxon, Budapest, Hungary).

### 4.2. Determination of Molar Absorptivity and Labeling Ratio

The optical properties, including absorbance and fluorescence spectra of fluorophores, were measured using a UV-Vis-NIR spectrophotometer (USB-ISS-UV/VIS, Ocean Optics, Dunedin, FL, USA). To measure molar extinction absorptivity, six samples of HSA-ZW800-PEG were prepared in saline, with final concentrations ranging from 0.78–25 µM. Aliquots of the working standard solution were transferred into a series of 10 mL Fischer brand disposable tubes. Subsequently, 200 µL of each sample was dispensed onto 96-well plates, and molar absorbance and fluorescence were measured using Cytation5. After thorough mixing, the absorbance spectrum of each sample was recorded on 96-well plates. The absorbance at the wavelength of maximum absorbance was then plotted against the final dye concentration (mM) to generate the calibration graph. The labeling ratio of ZW800-PEG to HSA was calculated using the extinction coefficients of HSA (35,700 M^−1^cm^−1^ at 280 nm) and ZW800-PEG (192,000 M^−1^cm^−1^ at 787 nm, correction factor 0.047 in water).

### 4.3. Photostability Assay Using NIR Light

To investigate the relationship between photostability and concentration, 10 mM stock solutions were diluted to obtain 100 µM working solutions in 200 µL of saline. These working solutions, with concentrations of 100, 50, 20, and 10 µM in saline, were then dispensed into a black 96-well plate. Subsequently, the 200 µL working solutions were exposed to continuous irradiation using a 760 nm laser diode at 4 mW/cm^2^, with white light (400–650 nm) at 40,000 lux and imaged at 30 min intervals up to 240 min. The fluorescence intensity signal (%) was calculated by analyzing the measured values with ImageJ version 1.53t using images captured at each time point. The data, presented as mean ± SD, assessed stability by measuring regions of interest (ROI) and comparing fluorescence intensity against the initial fluorescence signal.

### 4.4. In Vivo Biodistribution and Pharmacokinetics

The animals used in this study were housed in an AAALAC-certified facility and were treated in accordance with the institutional protocol #2016N000136 approved by the Massachusetts General Hospital Institutional Animal Care and Use Committee (MGH IACUC). Male CD-1 mice, aged 6–8 weeks and weighing 25–30 g, were procured from Charles River Laboratories (Wilmington, MA, USA). The intravenous injection of fluorophores was carried out at 50 nmol in saline. Blood samples were taken at 1, 3, 5, 10, 30, 60, 120, 180, and 240 min post-injection. To determine the pharmacokinetics including the distribution (t_1/2α_) and elimination (t_1/2β_) half-life values, the fluorescence intensities of serum samples in capillary tubes were evaluated under the NIR imaging system (*n* = 3 for each group). The biodistribution and clearance were evaluated 4 h post-injection of fluorophores, and major organs, including the heart, lung, liver, pancreas, spleen, kidney, duodenum, intestine, and muscle, were resected after intraoperative imaging. ImageJ version 1.53t was utilized to measure the fluorescence and background intensities of a region of interest (ROI) across each tissue. SBR = fluorescence/background was the formula used to compute the signal-to-background ratio (SBR). The results were shown as a bi-exponential decay curve using Prism software version 9.2 (GraphPad, San Diego, CA, USA).

### 4.5. Cellular Uptake and Viability Tests for ZW800-PEG and HSA-ZW800-PEG

NIH3T3 fibroblast cells and LLC lung cancer cells were obtained from ATCC (Manassas, VA, USA) and cultured in DMEM supplemented with 10% FBS and 1% penicillin/streptomycin in a humidified incubator at 37 °C before conducting experiments. For the in vitro cellular uptake study, LLC and NIH3T3 cells were seeded in 24-well plates at a density of 1 × 10^4^ and 7 × 10^3^ cells, respectively. After incubation for 48 h, the cells were treated with 2.5–100 µM of ZW800-PEG and HSA-ZW800-PEG for 3, 8, and 24 h. Subsequently, the cells were washed with fresh media three times, then both bright-field and fluorescence images were captured by Cytation5 (BioTek, Winooski, VT, USA). Fluorescence intensity images of each cell were measured using ImageJ version 1.53t. Cell viability (%) was calculated using the number of cells treated with 2.5–100 µM of HSA-ZW800-PEG, as shown in bright-field images, divided by the number of cells in the 2.5 µM group. The positive-to-negative ratio (PNR) values were calculated by dividing the average fluorescence intensity in LLC cells by that in NIH3T3 cells. Quantitative imaging analysis utilized four individual areas of images for accurate assessments.

### 4.6. In Vitro Inhibition Study

LLC and NIH3T3 cells were seeded in 24-well plates at a density of 1 × 10^4^ and 7 × 10^3^ cells, respectively. To determine the role of endocytosis in HSA-ZW800-PEG uptake, the cells were pretreated with 30 μM of Dyngo 4a (dynamin-dependent endocytosis inhibitor, Abcam, Waltham, MA, USA, No. 120689) for 30 min. The cells were then incubated with 50 μM of HSA-ZW800-PEG at 37 °C for 8 h. After washing twice with HBSS buffer, fluorescence images were captured using Cytation5.

### 4.7. Tumor-Bearing Mouse Models

To establish various tumor-bearing mouse models, including lung carcinoma, B6 male mice (Taconic Farms, Germantown, NY, USA) were subcutaneously injected with 2 × 10^5^ LLC cells suspended in DMEM/Matrigel (100 μL, 50 *v*/*v*%) in the flank region. Before the experiment, tumors were allowed to reach a size of approximately 100 mm^3^. Once tumor volumes reached around 100 mm^3^, mice were intravenously injected with 100 μL of HSA-ZW800-PEG (50 nmol, 2 μmol/kg) and ZW800-PEG (100 nmol, 4 μmol/kg). After 8 h for ZW800-PEG and 24 h for HSA-ZW800-PEG treatment, the mice were sacrificed, and tumor images were captured (*n* = 2, mean ± SD). Tumor-to-background ratio (TBR) signal ratios of the tumors compared with muscles or system background for in vivo NIR imaging were obtained at each time point.

### 4.8. Quantitative and Statistical Analysis

The fluorescence and background intensities of regions of interest (ROI) were quantified at each time point using customized FLARE imaging software and ImageJ version 1.53t. The SBR, TBR, and PNR were calculated using ImageJ as *I_ROI_*_/_*I_Auto_*, where $I_{ROI}$ denotes the average intensity of an ROI and $I_{Auto}$ represents the signal intensity of neighboring tissues, such as muscle or skin, obtained over the imaging period or system background. All NIR fluorescence images for a particular fluorophore were normalized identically for all conditions of an experiment. All data depict the mean ± standard deviation (SD) with at least 3 biological replicates. Statistical analysis was carried out using the unpaired Student’s *t*-test or one-way analysis of variance (ANOVA). Results are presented as mean standard deviation, and curve fitting was performed using Prism software version 9.2 (GraphPad, San Diego, CA, USA). We considered *p* values less than 0.05 significant: * *p* < 0.05, ** *p* < 0.01, *** *p* < 0.001, and **** *p* < 0.0001.

## 5. Conclusions

The streamlined synthesis and characterization of HSA-ZW800-PEG highlight its suitability for preclinical and potential clinical applications. The efficient and scalable production process of ZW800-PEG and its HSA conjugates ensures high yield and purity. The conjugation of HSA with ZW800-PEG notably improves chemical, physical, and optical stability under physiological conditions, addressing challenges encountered with ZW800-PEG in biological applications. Furthermore, the improved molecular weight and extinction coefficient enhance the in vivo performance of HSA-ZW800-PEG. The distinct distribution and elimination dynamics, coupled with the significantly longer blood half-life of HSA-ZW800-PEG, enhanced tumor targetability in both subcutaneous and orthotopic xenograft tumor-bearing animal models. It not only influences the pharmacokinetic profile, affecting retention time and clearance patterns but also enhances bioavailability for targeting tissues. These findings shed light on the intricate interplay between molecular design and pharmacokinetic behavior, guiding further development and optimization of targeted imaging agents and drug-delivery systems.

Nevertheless, certain challenges and limitations were identified. First, the specificity of HSA-ZW800-PEG in non-cancerous tissues requires further investigation. Understanding potential off-target interactions and uptake in normal tissues is crucial for ensuring the safety and precision of the approach. Additionally, while HSA-ZW800-PEG exhibits remarkable stability after a 4 h irradiation period, an extended evaluation over more prolonged periods could provide insights into its long-term stability under various physiological conditions.

For clinical translation, rigorous evaluation of safety and efficacy in human subjects is deemed crucial. Beyond its applications in imaging, the unique properties of HSA-ZW800-PEG position it as a promising candidate for drug-delivery systems. Exploring potential combinations with other imaging modalities, including PET, SPECT, CT, and MRI, holds the potential to enhance diagnostic capabilities. Moreover, extending evaluations of HSA-ZW800-PEG to a broader spectrum of cancer types will provide valuable insights into its versatility and potential applications across different malignancies.

In conclusion, while the study underscores the promising attributes of HSA-ZW800-PEG, addressing the outlined limitations and exploring future avenues will be pivotal for realizing its full potential in advancing cancer imaging and therapeutic strategies. Continued research and development will refine its applications and contribute to its broader impact in the field of cancer theranostics.

## Figures and Tables

**Figure 1 ijms-25-00559-f001:**
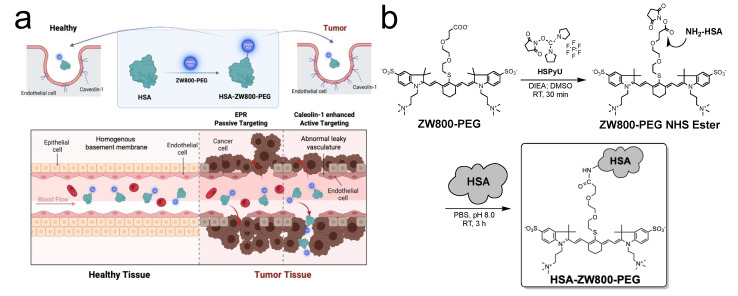
Preparation and optophysical analyses of HSA-ZW800-PEG. (**a**) Schematic illustration of passive and active tumor targeting using HSA-ZW800-PEG. (**b**) Chemical synthesis of HSA-ZW800-PEG. (**c**) High-performance liquid chromatography (HPLC) of ZW800-PEG and HSA-ZW800-PEG equipped with a photodiode array (PDA) detector (200–800 nm). 1x PBS, pH 7.4 was used as a running buffer. (**d**) Absorbance (abs) and fluorescence (Fl) spectra of HSA-ZW800-PEG at a concentration of 2.5 µM in 5% bovine serum albumin (BSA) solution. (**e**) Optophysical properties of HSA-ZW800-PEG by comparing with ZW800-PEG. MW = molecular weight; HD = hydrodynamic diameter; λ_Abs_ = absorption maxima; λ_Em_ = emission maxima; ε = molar extinction coefficient; QY = quantum yield. * Values were adapted from our previous publication [19].

**Figure 2 ijms-25-00559-f002:**
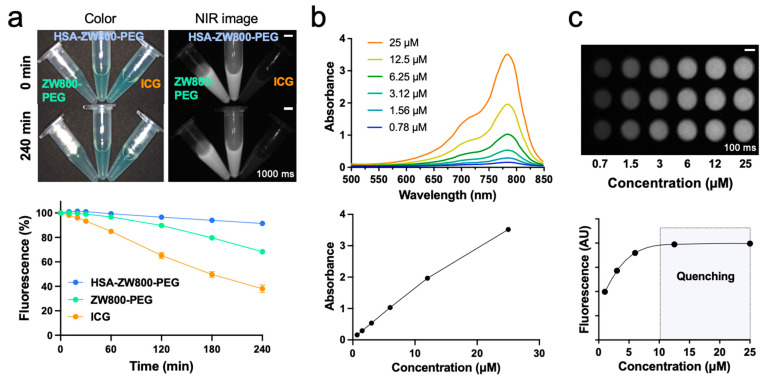
In vitro optophysical stability of HSA-ZW800-PEG. (**a**) Color and NIR images (exposure time = 1000 ms, Scale bars = 0.5 cm) of ZW800-PEG, HSA-ZW800-PEG, and ICG in saline at a concentration of 20 µM pre- and post-exposure 760 nm laser diode (3 mW·cm^−2^) with a white light up to 240 min. The photostability pattern of each fluorophore was calculated by plotting the percent fluorescence (*n* = 3; mean ± SD). (**b**) Absorbance spectra of HSA-ZW800-PEG at various concentrations in saline. (**c**) NIR fluorescence images of HSA-ZW800-PEG at different concentrations in saline (exposure time = 100 ms, Scale bars = 0.5 cm). Optical fluorescence quenching of HSA-ZW800-PEG was observed at concentrations over 12.5 µM.

**Figure 3 ijms-25-00559-f003:**
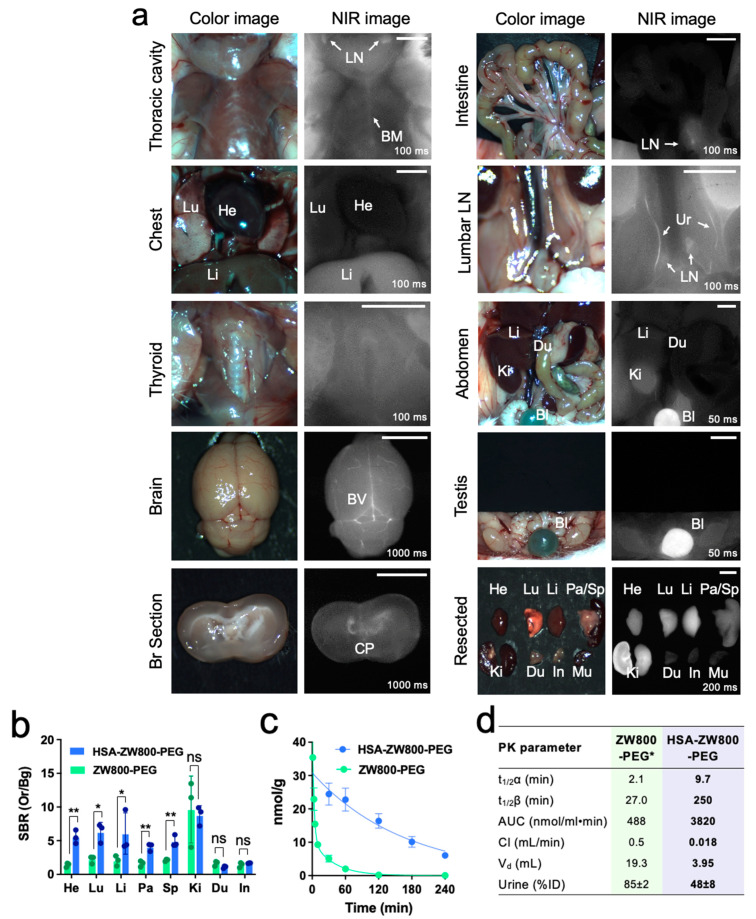
Biodistribution and pharmacokinetics of HSA-ZW800-PEG. (**a**) 50 nmol of HSA-ZW800-PEG was injected retro-orbitally in CD-1 mice at 4 h prior to imaging. Abbreviations used are Bl: bladder; BM: bone marrow; BV: brain vasculature; CP: choroid plexus; Du: duodenum; In: intestine; Ki: kidneys; Li: liver; LN: lymph nodes; Lu: lungs; Pa: pancreas; Sp: spleen; Ur: ureter. Scale bars = 0.5 cm. (**b**) Signal-to-background ratio (SBR) of resected organs (Ors) against system background (Bg) measured by ImageJ version 1.53t and compared statistically using unpaired multiple *t*-tests (*n* = 3; mean ± SD; * *p* < 0.05, ** *p* < 0.01, ns = not significant). (**c**) Blood curves of HSA-ZW800-PEG and ZW800-PEG in normal mice. (**d**) Pharmacokinetic (PK) parameters of fluorophores (t_1/2α_: distribution half-life; t_1/2β_: elimination half-life; AUC: area under the curve; V_d_: volume of distribution; % ID: percent injected dose) were calculated using Prism 9 software (*n* = 3, mean ± SD). Urinary excretion was calculated at 4 h post-injection. * Values were adapted from our previous publication [19].

**Figure 4 ijms-25-00559-f004:**
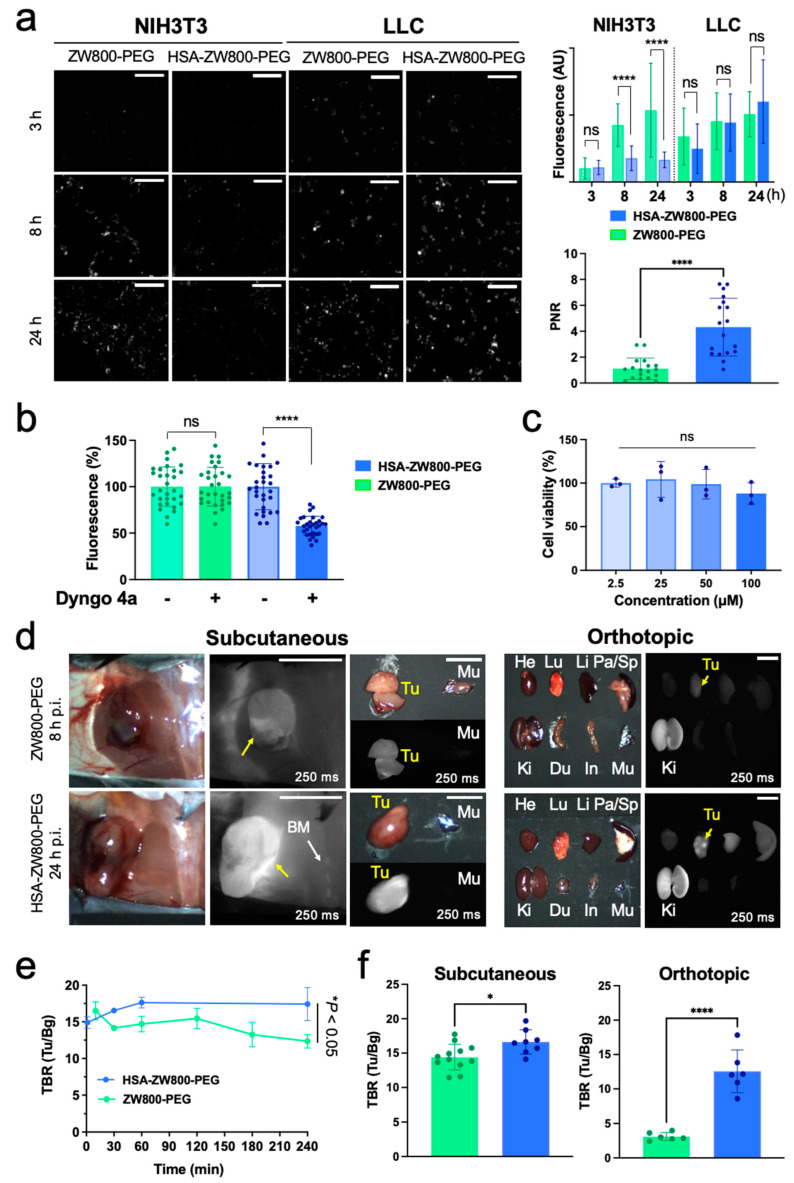
Tumor targetability of HSA-ZW800-PEG and ZW800-PEG in lung cancer cell lines and tumor-bearing animal models. (**a**) Cellular uptake and intracellular localization after treating 100 µM of each fluorophore in NIH3T3 and LLC cell lines. Fluorescence images were obtained using Cytation5 at each time point and compared statistically using unpaired multiple *t*-tests (*n* = 20–30, mean ± SD; **** *p* < 0.0001, ns = not significant). The positive-to-negative ratio (PNR) was analyzed identically for the experiment condition by analyzing NIR images using ImageJ version 1.53t. Scale bars = 100 μm. (**b**) Cell binding inhibition test after pretreatment with 30 μM of Dyngo 4a followed by incubation with 100 μM of HSA-ZW800-PEG and ZW800-PEG for 8 h. Statistically compared using unpaired *t*-tests (*n* = 30, mean ± SD; **** *p* < 0.0001, ns = not significant) (**c**) Cell viability test was performed by treating 2.5–100 µM of HSA-ZW800-PEG in NIH3T3 cell lines for 24 h. The number of cells in different concentrations was calculated using bright-field images and compared statistically using one-way ANOVA followed by Tukey’s multiple comparisons test (*n* = 3, ns = not significant). (**d**) 50–100 nmol (2–4 μmol/kg) of each fluorophore was injected into orthotopic and subcutaneous xenograft LLC tumor-bearing B6 male mice. Shown are representative images obtained at 8 h post-injection of ZW800-PEG and 24 h post-injection of HSA-ZW800-PEG, respectively. Abbreviations used are BM: bone marrow; Du: duodenum; In: intestine; Ki: kidneys; Li: liver; Lu: lungs; Pa: pancreas; Sp: spleen; Tu: tumor. Scale bars = 0.5 cm. (**e**) Kinetics of tumor-to-background ratios (TBRs) obtained from subcutaneous tumors (Tu) compared with background (Bg). (**f**) TBR of obtained tumors from subcutaneous versus orthotopic xenograft animal models. NIR fluorescence images for each condition have identical exposure times (exposure time = 250 ms). Fluorescence images were obtained using the FLARE imaging system at each time point and compared statistically using unpaired *t*-tests (*n* = 6 or 12; mean ± SD; * *p* < 0.05, **** *p* < 0.0001).

## Data Availability

All data are available in the main text.

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
