# Peer review of "HSA-ZW800-PEG for Enhanced Optophysical Stability and Tumor Targeting"

_ijms, 2023, doi:10.3390/ijms25010559_

Round 1
Reviewer 1 Report
Comments and Suggestions for Authors
This manuscript by Choi et al reports the development of HSA800-PEG for enhanced FL imaging in vivo and in vitro, and enhanced tumor targeting. The HSA and ZW800-PEG conjugate offers optimized NIR imaging nanoprobe for in vivo imaging and tumor targeting, which shows high potential for clinical translation. This work is interesting and could be another important work to the related field.
Specific comments:
1. The concept for PEG is not correct in the work. Polyethylene glycols (PEGs) have long molecular chains with high molecular weights, while ZW800-PEG only has one repeating unit, which often is termed as oligo(ethylene glycol) (OEG). ZW800-PEG and HSA800-PEG should be ZW800-OEG and HSA800-OEG.
2. The doses for HSA800-PEG and ZW800-PEG are 50 nmol (2 μmol/kg) and 100 nmol (4 μmol/kg), respectively. The dose for ZW800 dye is around 100 nmol per mice, which is very high for live mice imaging. Why is the dose so high? What will happen if the dose decreases to 10 nmol for example? Does the in vivo performance dose-dependent?
3. The size and zeta potential for HSA800-PEG might be characterized by DLS except for GPC.
4. The importance of this work is the enhanced photo stability of ZW800 after conjugation to HSA. The discussion part can further discuss the factors that impact dye FL imaging in vivo and the related applications. Some references might be useful in the discussion part for references [doi.org/10.1021/acsnano.9b06504; doi.org/10.1021/acsnano.1c07010; doi.org/10.1016/j.biomaterials.2022.121824].
Author Response
- The concept for PEG is not correct in the work. Polyethylene glycols (PEGs) have long molecular chains with high molecular weights, while ZW800-PEG only has one repeating unit, which often is termed as oligo(ethylene glycol) (OEG). ZW800-PEG and HSA800-PEG should be ZW800-OEG and HSA800-OEG.
We appreciate the reviewer’s valuable suggestion. As pointed out, the three repeating ethylene glycol units could be considered origo-ethylene glycol (OEG). However, we initially made this nomenclature “ZW800-PEG” to show the unique properties of the new fluorophore (Yang et al. Angew Chem Int Ed Engl 2021). To avoid confusion, although we appreciate the reviewer’s suggestion, we would like to use the same naming for our future publications. Also, according to Reviewer #3’s comments, we changed the abbreviation from HSA800-PEG to HSA-ZW800-PEG.
[19] Yang, C.; Wang, H.; Yokomizo, S.; Hickey, M.; Chang, H.; Kang, H.; Fukuda, T.; Song, M. Y.; Lee, S. Y.; Park, J. W.; Bao, K.; Choi, H. S., ZW800-PEG: A Renal Clearable Zwitterionic Near-Infrared Fluorophore for Potential Clinical Translation. Angew Chem Int Ed Engl 2021, 60, (25), 13847-13852.
- The doses for HSA800-PEG and ZW800-PEG are 50 nmol (2 mol/kg) and 100 nmol (4 μmol/kg), respectively. The dose for ZW800 dye is around 100 nmol per mice, which is very high for live mice imaging. Why is the dose so high? What will happen if the dose decreases to 10 nmol for example? Does the in vivo performance dose-dependent?
Fig. R1. Tumor targeting of cRGD-ZW800-PEG in LLC-bearing animal models. 10-100 nmol of cRGD-ZW800-PEG was injected intravenously 4 h prior to imaging. |
This is a very important question. The standard injection dose of indocyanine green (FDA-approved NIR dye) is 100 nmol (25 mg human-equivalent dose) [R1-3]. Generally, we inject a lower dose than the standard dose, but it should depend on the brightness of the tested dyes as well as their toxicity. Since HSA-ZW800-PEG is non-toxic up to 100 µM concentration (Fig. 2e), we could increase the injection dose accordingly. Also, in our previous study [R4], we tested the dose-dependent tumor targeting of cRGD-ZW800-PEG (cRGD-conjugated ZW800-PEG), where 50 nmol was found to be optimal (see Fig. R1). The TBR (tumor-to-background ratio) increased dose-dependently while no difference was found between 50 and 100 nmol. To be a comparable dose with the best condition for tumor imaging, we chose 50 nmol for HSA800-PEG. Since two ZW800-PEG moieties were conjugated on an HSA800-PEG (labeling ratio = 2.1, Fig. 1e), the dose of ZW800-PEG was decided to be 100 nmol, the same dye dose. We added this info on page 6, lines 180-181. “Since two ZW800-PEG moieties are conjugated on the HSA surface (labeling ratio = 2.1), different injection doses were applied.”
R1) Reagan-Shaw S et al. Dose Translation from animal to human studies revisited. FASEB J. 2008, 22, 659–661.
R2) Kelley et al. Body surface area of female swine. J Anim Sci. 1973, 36 (5): 927-930.
R3) Center for Drug Evaluation and Research, Center for Biologics Evaluation and Research. (2002) Estimating the safe starting dose in clinical trials for therapeutics in adult healthy volunteers, U.S. Food and Drug Administration, Rockville, Maryland, USA.
R4) Bao K et al. Ultralow background near‐infrared fluorophores with dual‐channel intraoperative imaging capability. Adv Healthc Mater 2023, 12, (12), 2203134.
- The size and zeta potential for HSA800-PEG might be characterized by DLS except for GPC.
This is another important point. In principle, size exclusion chromatography (SEC)-based hydrodynamic diameter measurements, including GPC and GFC, are very similar to dynamic light scattering (DLS), where autocorrelation is carried out on a varying light signal to compensate for fluctuations induced by the Brownian motion. DLS functions as a supplemental, non-destructive method that finds high molecular weight aggregates in their natural habitat, whereas the SEC method may miss some aggregates. HSA samples keep the intact structure after conjugating dyes on the surface, we don’t expect any high molecular weight aggregates. We agree that SEC alone could potentially be misleading, but we previously confirmed that DLS and GFC values are similar for glomerular structured nanoparticles (Fig. R2) [Choi et al. Nat Biotechnol 2007]. We also added two more references in the main manuscript to support our data [41-43].
Fig. R2. Measurement of hydrodynamic diameter and interaction of the organic coating with serum proteins. The emission wavelength of cysteine-coated quantum dots (QD-Cys) increased with the size of core/shell, measured by transmission electron microscopy (TEM), dynamic light scattering (DLS), and gel filtration chromatography (GFC). TEM size data for each sample were determined from the average of Z150 measurements. DLS and GFC measurements (mean ± s.d.) were from N=3 independent experiments. %PD, Polydispersity.
- Kummari, R.; Bose, K., Gel Filtration Chromatography. In 2022; pp 199-219.
- Choi, H. S.; Liu, W.; Misra, P.; Tanaka, E.; Zimmer, J. P.; Itty Ipe, B.; Bawendi, M. G.; Frangioni, J. V., Renal clearance of quantum dots. Nat Biotechnol 2007, 25, (10), 1165-70.
- Al-Ghobashy, M. A.; Mostafa, M. M.; Abed, H. S.; Fathalla, F. A.; Salem, M. Y., Correlation between Dynamic Light Scattering and Size Exclusion High Performance Liquid Chromatography for Monitoring the Effect of pH on Stability of Biopharmaceuticals. Journal of Chromatography B 2017, 1060, 1-9.
- The importance of this work is the enhanced photostability of ZW800 after conjugation to HSA. The discussion part can further discuss the factors that impact dye FL imaging in vivo and the related applications. Some references might be useful in the discussion part for references [doi.org/10.1021/acsnano.9b06504; doi.org/10.1021/acsnano.1c07010; doi.org/10.1016/j.biomaterials.2022.121824].
We apologize for missing such important references. We now added the suggested references accordingly in the main text.
- Deng, H.; Li, X.; Ju, J.; Mo, X.; Ge, G.; Zhu, X., Multifunctional nanoprobes for macrophage imaging. Biomaterials 2022, 290, 121824.
- Deng, H.; Konopka, C. J.; Prabhu, S.; Sarkar, S.; Medina, N. G.; Fayyaz, M.; Arogundade, O. H.; Vidana Gamage, H. E.; Shahoei, S. H.; Nall, D.; Youn, Y.; Dobrucka, I. T.; Audu, C. O.; Joshi, A.; Melvin, W. J.; Gallagher, K. A.; Selvin, P. R.; Nelson, E. R.; Dobrucki, L. W.; Swanson, K. S.; Smith, A. M., Dextran-Mimetic Quantum Dots for Multimodal Macrophage Imaging In Vivo, Ex Vivo, and In Situ. ACS Nano 2022, 16, (2), 1999-2012.
- Deng, H.; Konopka, C. J.; Cross, T.-W. L.; Swanson, K. S.; Dobrucki, L. W.; Smith, A. M., Multimodal Nanocarrier Probes Reveal Superior Biodistribution Quantification by Isotopic Analysis over Fluorescence. ACS Nano 2020, 14, (1), 509-523.
Please see the enclosed revision cover letter for more details.

Reviewer 2 Report
Comments and Suggestions for Authors
The authors of the work article entitled “HSA800-PEG for Enhanced Optophysical Stability and Tumor Targeting” propose the conjugation of the near-infrared fluorophore ZW800-PEG to human serum albumin, eventually addressing some of the problems that small molecules face in biological applications.
As it is, the article is very well written, in a manner that the reader well perceives. The results are well addressed and explained. However, there is still some key factor missing:
1. Since the authors say that the conjugate produced is zwitterionic, I would like to see some of this evidence, for example the change in charge at different pH, since the surface charge is significant in systemic administration.
2. I would also like to see some comparison with other small molecules, either with ones already in use or others in the literature, namely comparing the photostability, optical quenching, and molar extinction coefficient, since these are key factors for bioimaging.
3. In general, I would like more comparison with works in the literature, to give the readers some way of judgment, specifically with the biodistribution/in vivo assays.
4. There seems to be missing a “[REF]” in page 4 line 108.
5. The authors should also verify the irradiation parameters, since on page 4, line 109 it says, “4 mW/cm2” and there is no mention of this in the rest of the manuscript.
6. I would like to see some concrete future perspectives and for the authors to comment on the potential of this small molecule in the therapeutic area.
Author Response
The authors of the work article entitled “HSA800-PEG for Enhanced Optophysical Stability and Tumor Targeting” propose the conjugation of the near-infrared fluorophore ZW800-PEG to human serum albumin, eventually addressing some of the problems that small molecules face in biological applications.
As it is, the article is very well written, in a manner that the reader well perceives. The results are well addressed and explained. However, there is still some key factor missing:
- Since the authors say that the conjugate produced is zwitterionic, I would like to see some of this evidence, for example the change in charge at different pH, since the surface charge is significant in systemic administration.
We apologize for the confusion and appreciate the reviewer’s comments. As described in our manuscript, zwitterionic ZW800-PEG (+3/-3, [19] Yang et al. Angew Chem Int Ed Engl 2021) was used to conjugate on HSA. The changes in charge status of ZW800-PEG were calculated by MarvinSketch (Fig. R3). Interestingly, no isoelectric point was found for ZW800-PEG, which is zwitterionic in the physiological conditions and partially protonated under pH 6.5. The two sulfonates and two quats on both sides of the indole rings maintain geometrically balanced surface charges on the fluorophore moiety, which is the key to lower background tissue uptake (Angew. Chem. Int. Ed. 2011, 50, 6258; Mol. Imaging Biol. 2016, 18, 52; Nat. Biotechnol. 2013, 31, 148). We previously investigated the influence of surface charges for many fluorophores (Adv. Photonics Res. 2023, 4, 2300036; Nature Nanotechnology 2010, 5, 42), where minimum to no effects were found in zwitterionic molecules during systemic circulations.
Fig. R3. Isoelectric point measurement of ZW800-PEG using MarvinSketch (ChemAxon, Budapest, Hungary).
- I would also like to see some comparison with other small molecules, either with ones already in use or others in the literature, namely comparing the photostability, optical quenching, and molar extinction coefficient, since these are key factors for bioimaging.
We appreciate the reviewer for this important question. We compared the optical properties of HSA-ZW800-PEG with already published two representative fluorophores, ZW800-PEG and ICG (Fig. 1e and Fig. 2). We added the extinction coefficients and quantum yields of HSA-ZW800-PEG which was similar to the fluorophore itself (Fig. 1e).
Figure 1. Preparation and optophysical analyses of HSA-ZW800-PEG. (e) Optophysical properties of HSA-ZW800-PEG by comparing with ZW800-PEG. MW = molecular weight; HD = hydrodynamic diameter; lAbs = absorption maxima; lEm = emission maxima; ε = molar extinction coefficient; QY = quantum yield. *Values were adapted from our previous publication [19].
The optical properties including photostability (Fig. 2a,b) and optical quenching (Fig. 2c) were added in the main text accordingly.
Figure 2. In vitro optophysical stability of HSA-ZW800-PEG. (a) Color and NIR images (exposure time = 1,000 ms) of ZW800-PEG, HSA-ZW800-PEG, and ICG in saline at a concentration of 20 µM pre- and post-exposure 760 nm laser diode (3 mW×cm-2) with a white light up to 240 min. The photostability pattern of each fluorophore was calculated by plotting the percent fluorescence (n = 3; mean ± SD). (b) Absorbance spectra of HSA-ZW800-PEG at various concentrations in saline. (c) NIR fluorescence image of HSA-ZW800-PEG at different concentrations in saline (exposure time = 100 ms). Optical fluorescence quenching of HSA-ZW800-PEG was observed at concentrations over 12 µM.
- In general, I would like more comparison with works in the literature, to give the readers some way of judgment, specifically with the biodistribution/in vivo assays.
We appreciate the reviewer’s comments. We added more references to show biodistribution and in vivo performances of NIR fluorophores accordingly.
- There seems to be missing a “[REF]” in page 4 line 108.
We apologize for the mistake. We’ve added an appropriate reference on page 4, line 112.
“[42] James, S.; Neuhaus, K.; Murphy, M.; Leahy, M., Contrast agents for photoacoustic imaging: a review of stem cell tracking. Stem Cell Research & Therapy 2021, 12, (1), 511.”
- The authors should also verify the irradiation parameters, since on page 4, line 109 it says, “4 mW/cm2” and there is no mention of this in the rest of the manuscript.
We apologize for not providing detailed information. We added more details on page 10, lines 334-336. “Subsequently, the 200 µL working solutions were exposed to continuous irradiation using a 760 nm laser diode at 4 mW/cm², with white light (400–650 nm) at 40,000 lux and imaged at 30 min intervals up to 240 min.”
Please see the enclosed revision cover letter for more details.

Reviewer 3 Report
Comments and Suggestions for Authors
The authors presented the paper "HSA800-PEG for Enhanced Optophysical Stability and Tumor Targeting"
Thank you for an exciting paper with mouse experiments. However, I have some comments.
1) I recommend changing abbreviation HSA800-PEG into HSA-PEG-ZW800 (or smth. like this) to not confuse readers what HSA800 means. Moreover, It looks like that it is just albumin PEG complex.
2) Introduction. I recommend decryption why albumin so specific for cancer tissue like EPR effect, receptor interaction with some references.
3) About the construction design. Please, enlarge the discussion why covalent approach is better in your opinion than noncovalent binding. See the work: https://www.sciencedirect.com/org/science/article/abs/pii/S1043180221059905
10.1021/acs.bioconjchem.9b00760
Please, add the explanation in the Introduction and Discussion section. It will be exciting, of course, to see what is better using your dye.
4) For the sections 2.1-2.2. Not only wavelength is essential for fluorescence imaging but quantum yield. Extinction coefficient may show perspectives but is not the same. For example, a fluorescein have not so high extinction but quantum yield of fluorescence of 80%. Have you studied quantum yield on free dye and your conjugate?
5) Have you studied the toxicity on cell cultures? I don't see clear standard procedures in the paper. I understand that it should be safe but in some cases nonspecific albumin acylation by active esters can result in non so safe conjugate. Cytotoxicity is a mandatory procedure before animal experiments.
6) It will be excellent to present in the Conclusion section limitations and future outlooks of the work.
line 61 better 19 days
Comments on the Quality of English Language
Minor editing of English language required
Author Response
The authors presented the paper "HSA800-PEG for Enhanced Optophysical Stability and Tumor Targeting" Thank you for an exciting paper with mouse experiments. However, I have some comments.
- I recommend changing abbreviation HSA800-PEG into HSA-PEG-ZW800 (or smth. like this) to not confuse readers what HSA800 means. Moreover, It looks like that it is just albumin PEG complex.
HSA800-PEG is a bioconjugate between human serum albumin (HSA) and a zwitterionic fluorophore named ZW800-PEG (Yang et al. Angew Chem Int Ed Engl 2021). To avoid any confusion, we revised the nomenclature of HSA800-PEG to HSA-ZW800-PEG. We appreciate the reviewer’s suggestion.
- I recommend decryption why albumin so specific for cancer tissue like EPR effect, receptor interaction with some references.
This is a very important point. We now clearly explained the tumor targeting mechanism of HSA-ZW800-PEG in the discussion section on page 8, lines 250-256: “This aligns with numerous reports on the longer blood half-life and passive targeting of HSA-conjugates through the enhanced permeability and retention (EPR) effect [49-51]. Additionally, the high expression of caveolin-1 in tumor stroma, a significant component of caveolae, has been associated with improved responses to albumin-conjugated drugs in non-small cell lung cancer (NSCLC) patients [20,52]. Similar results were found when Cy5.5 was conjugated on albumin, where tumor targeting was improved but background uptake was also increased [53].”
- About the construction design. Please, enlarge the discussion why covalent approach is better in your opinion than noncovalent binding. See the work: https://www.sciencedirect.com/org/science/article/abs/pii/S104318022105990510.1021/acs.bioconjchem.9b00760. Please, add the explanation in the Introduction and Discussion section. It will be exciting, of course, to see what is better using your dye.
We appreciate the reviewer for this important suggestion. The suggested reference mentions that interaction modification between dyes and serum albumin regulates the tumor targetability but the background signal also increases. In addition to the dye accumulation in tumors, the lower background signal from other tissues is also very important for tumor detection and visualization because what we observe is not just the signal itself but the signal-to-background ratio (https://journals.sagepub.com/doi/full/10.2310/7290.2009.00033). Furthermore, we also previously investigated the effects of serum albumin on the tumor targetability of NIR fluorophores [R5].
[R5] Syed Muhammad Usama, G. Kate Park, Shinsuke Nomura, Yoonji Baek, Hak Soo Choi, Kevin Burgess, Role of Albumin in Accumulation and Persistence of Tumor Seeking Cyanine Dyes. 2020, Bioconjug. Chem., 31(2), 248-259.
Physically adsorbed fluorophores like ICG:HSA are limited for tumor targeting due to the following reasons: 1) Beyond a 1:1 molar ratio of ICG to HSA, the improvement in total fluorescence yield becomes saturated, potentially allowing dissociated ICG to travel freely in the body [36,37] and 2) the binding pocket of HSA can also be obstructed by ICG, leading to quick dissociation of ICG after passage through the liver [38].
Overall, albumin-conjugated fluorophores improve the TBR significantly and we added the following sentences in the revision on page 8: “This aligns with numerous reports on the longer blood half-life and passive targeting of HSA-conjugates through the enhanced permeability and retention (EPR) effect [49-51]. Additionally, the high expression of caveolin-1 in tumor stroma, a significant component of caveolae, has been associated with improved responses to albumin-conjugated drugs in non-small cell lung cancer (NSCLC) patients [20,52]. Similar results were found when Cy5.5 was conjugated on albumin, where tumor targeting was improved but background uptake was also increased [53].”
[53] Wooram Um, Jooho Park, Ahye Youn, Hanhee Cho, Seungho Lim, Jong Won Lee, Hong Yeol Yoon, Dong-Kwon Lim, Jae Hyung Park, and Kwangmeyung Kim, A Comparative Study on Albumin-Binding Molecules for Targeted Tumor Delivery through Covalent and Noncovalent Approach. 2019, Bioconjug. Chem., 30, 3107-3118.
- For the sections 2.1-2.2. Not only wavelength is essential for fluorescence imaging but quantum yield. Extinction coefficient may show perspectives but is not the same. For example, a fluorescein have not so high extinction but quantum yield of fluorescence of 80%. Have you studied quantum yield on free dye and your conjugate?
We added quantum yield measurements for HSA800-PEG and ZW800-PEG in Figure 1e and an explanation on page 3, lines 93-94. We appreciate the reviewer’s comments.
- Have you studied the toxicity on cell cultures? I don't see clear standard procedures in the paper. I understand that it should be safe but in some cases nonspecific albumin acylation by active esters can result in non so safe conjugate. Cytotoxicity is a mandatory procedure before animal experiments.
This is a very important question. Cytotoxicity of ZW800-PEG was already evaluated at the maximum concentration of 100 μM in our previous report: “[19] Yang et al. ZW800-PEG: A Renal Clearable Zwitterionic Near-Infrared Fluorophore for Potential Clinical Translation. Angew Chem Int Ed Engl 2021, 60, (25), 13847-13852.” The result clearly confirmed that there is no significant cytotoxicity of ZW800-PEG at such a high concentration. Since HSA is known to be nontoxic, we also expected that the HSA-conjugated ZW800-PEG would not show any toxicity. According to the reviewer’s comment, however, we performed the cell viability test of HSA-ZW800-PEG and added it to Fig. 4c. As expected, HSA-ZW800-PEG did not show any cytotoxicity in NIH3T3 fibroblast cells in the concentration range of 1-100 µM. To explain this more clearly, we added the above information in Figure 4c and the Results section.
Figure 4c. Cell viability test was performed by treating 1-100 µM of HSA-ZW800-PEG in NIH3T3 cell lines. The number of cells in different concentrations was calculated using bright-field images and compared statistically using one-way ANOVA followed by Tukey’s multiple comparisons test (n = 3, ns = not significant). |
- It will be excellent to present in the Conclusion section limitations and future outlooks of the work. line 61 better 19 days.
This is an excellent suggestion. We added the following limitations and future outlooks in conclusions on page 9: “Nevertheless, certain challenges and limitations are identified. Firstly, the specificity of HSA-ZW800-PEG in non-cancerous tissues requires further investigation. Understanding potential off-target interactions and uptake in normal tissues is crucial for ensuring the safety and precision of the approach. Additionally, while HSA-ZW800-PEG exhibits remarkable stability after a 4 h irradiation period, an extended evaluation over more prolonged periods could provide insights into its long-term stability under various physiological conditions.
For clinical translation, rigorous evaluation of safety and efficacy in human subjects is deemed crucial. Beyond its applications in imaging, the unique properties of HSA-ZW800-PEG position it as a promising candidate for drug delivery systems. Exploring potential combinations with other imaging modalities, including CT, MRI, PET, and SPECT, holds the potential to enhance diagnostic capabilities. Moreover, extending evaluations of HSA-ZW800-PEG to a broader spectrum of cancer types will provide valuable insights into its versatility and potential applications across different malignancies.
In conclusion, while the study underscores the promising attributes of HSA-ZW800-PEG, addressing the outlined limitations and exploring future avenues will be pivotal for realizing its full potential in advancing cancer imaging and therapeutic strategies. Continued research and development will refine its applications and contribute to its broader impact in the field of cancer theranostics.”
We also corrected 19 d to 19 days on page 2 line 62 accordingly.
Please see the enclosed revision cover letter for more details.

Round 2
Reviewer 3 Report
Comments and Suggestions for Authors
Thank you for the revised version of the manuscript.
Comments on the Quality of English LanguageMinor editing of English language required